# A century of weekly notifiable disease incidence data by province in Canada

**David J. D. Earn**[1,2], **Gabrielle MacKinnon**[1,3,4,5], **Samara Manzin**[1,6],
**Michael Roswell**[1,7,8], **Steve Cygu**[1,9], **Chyunfung Shi**[10], **Benjamin M. Bolker**[1,10],
**Jonathan Dushoff**[2,10], **Steven C. Walker**[1]*

1 Department of Mathematics and Statistics, McMaster University, Hamilton, Ontario, Canada, 2 M. G.
DeGroote Institute for Infectious Disease Research, McMaster University, Hamilton, Ontario, Canada,
3 Department of Epidemiology, Biostatistics, and Occupational Health, McGill University, Montréal,
Québec, Canada, 4 Lady Davis Institute for Medical Research, Jewish General Hospital, Montréal,
Québec, Canada, 5 Department of Mathematics and Statistics, McGill University, Montréal, Québec,
Canada, 6 Department of Biology, McGill University, Montréal, Québec, Canada, 7 Department of
Biology, University of Maryland, College Park, Maryland, United States of America, 8 Department of
Entomology, University of Maryland, College Park, Maryland, United States of America, 9 African
Population Health and Research Center, Nairobi, Kenya, 10 Department of Biology, McMaster University,
Hamilton, Ontario, Canada

* swalk@mcmaster.ca

pgph.0005550

Hong Kong Li Ka Shing Faculty of
Medicine, HONG KONG

**Peer Review History:** PLOS recognizes
the benefits of transparency in the peer
review process; therefore, we enable the
publication of all of the content of peer
review and author responses alongside
final, published articles. The editorial
history of this article is available here:
https://doi.org/10.1371/journal.pgph.
0005550

## Abstract

Canadian notifiable disease surveillance programmes have recorded communicable
disease incidence data, dating back to the late 19th century. A Public Health Agency
of Canada web-portal provides summaries of these data from 1924–2023, but lacks
details on how incidence varies seasonally and geographically among provinces.
The sub-annual (weekly, monthly, quarterly) and sub-national (provincial, territo-
rial) data required to study such patterns appear in government documents, but are
available only in typewritten or handwritten hard copies. We digitized and collated
these sources to make sub-annual and sub-national Canadian disease incidence
data conveniently available for researchers. We manually transcribed hard copies
into digital spreadsheets resembling the originals, enabling accurate transcription
through easier cross-checking. We supplemented these historical data sources with
more recent digital spreadsheets obtained directly from two provincial agencies. We
standardized and combined these spreadsheets into consistent, machine-readable
CSV files containing 1,631,380 incidence values from 1903–2021. Because multi-
ple publications and agencies reproduced case counts from the same surveillance
system, and many publications reported the same cases at multiple levels of aggre-
gation, many cases were counted more than once among these 1,631,380 incidence
values. We reconciled overlapping counts to produce a dataset containing 934,010
unique incidence values at sub-national and sub-annual scales (829,689 weekly;
567 2-weekly; 82,267 monthly; 20,967 quarterly; 520 3-quarterly) covering 139
diseases stratified by province or territory. We illustrate the value of these sub-
annual and sub-national data using two examples: synchronized annual cycles of

**Data availability statement:** All data used in this manuscript are accessible through a public GitHub repository (https://github.com/canmod/iidda). This repository contains all of the scripts and source documents that we used to produce our cleaned datasets, which have been deposited in Zenodo (https://doi.org/10.5281/zenodo.17459492). All code used to produce the figures in the manuscript are available on another public GitHub repository (https://github.com/canmod/candid).

**Funding:** The Natural Sciences and Engineering Research Council of Canada (NSERC) has supported this work for 25 years through Discovery Grants to DJDE. In addition, the following NSERC grants and awards enabled us to bring the project to completion: 560516-2020 to the Canadian Network for Modelling Infectious Diseases, supporting DJDE, BMB, and JD; 2021-04068 to DJDE; 2016-05488 to BMB; 574054-2022 to GM. The funders had no role in study design, data collection and analysis, decision to publish, or preparation of the manuscript.

**Competing interests:** The authors have declared that no competing interests exist.

poliomyelitis across provinces and spatially heterogeneous resurgence of whooping cough in the 1990s. Canada's infectious disease surveillance has produced a detailed record of sub-annual and sub-national disease incidence data that remains largely unexplored. This record is now available as the Canadian Notifiable Disease Incidence Dataset (CANDID), hosted on a publicly accessible website along with code to reproduce it, and scans of the original sources.

## Introduction

Learning from data on past communicable disease outbreaks and recurrent epidemics is an important component of public health planning [1–5]. The year 2024 marked the 100th anniversary since the Canadian federal government began collecting such data through notifiable disease surveillance programmes [6–10]. Several provinces conducted surveillance before 1924 going back to the late 19th century, although here we focus on the period from 1903 to 2021. The Public Health Agency of Canada (PHAC) provides summaries of these data as annual, national totals since 1924 through an online portal (https://diseases.canada.ca/notifiable). These coarsely aggregated data are useful for understanding broad trends, but provide no information on seasonal patterns of incidence within years or on spatial patterns across provinces. For example, annual data cannot be used to estimate the timing and shape of an outbreak curve, and national data cannot be used to assess whether provinces displayed different patterns of spread in the vaccine era.

The Canadian notification system has at times collected more informative weekly, monthly, and quarterly incidence data, stratified by disease and province/territory, but these data are not available through the PHAC portal. We use the term sub-annual to refer to surveillance data reported at a finer temporal resolution than one year (e.g., weekly, monthly, or quarterly), as opposed to annual data aggregated over the full year. Likewise, we use the term sub-national to refer to data reported at a finer spatial resolution than the national level (e.g., by province or territory). These higher-resolution data enable investigation of variation within years and across regions, and can be aggregated to coarser temporal or spatial resolutions when broader comparisons are needed. The reverse, however, is not possible, making higher-resolution data inherently more versatile.

It has generally been prohibitively challenging for researchers to find and access these sub-annual and sub-national data. Still, we show here that much of this information exists in government publications (either as hard copy reference material or online), in data and documents obtained directly from government agencies, and in provincial publications, some of which predate the establishment of the federal system in 1924. This information appeared under a variety of evolving titles often issued by Statistics Canada (and its predecessor the Dominion Bureau of Statistics), Health Canada, and provincial and territorial health departments. The diversity of publication formats and naming conventions has made systematic retrieval challenging. To address this, we have documented the available resources in this literature (see Section A in S1 Appendix) and digitized the data they contain using

open-source tools that we developed. Here, we introduce CANDID (Canadian Disease Incidence Dataset), a curated dataset that integrates and cleans these disparate data sources to create a comprehensive and accessible digital record of sub-annual (weekly, monthly, quarterly) and sub-national (provincial, territorial) Canadian notifiable disease incidence data–including substantial amounts of data that were publicly available in principle but almost entirely overlooked. Our objective is to make these sub-annual and sub-national disease incidence data, which have been difficult to obtain, conveniently available for public health and research use.

With this paper we announce the existence of CANDID on a publicly available web site (Section C in S1 Appendix) and illustrate its value. Two examples illustrate the advantages of the sub-annual and sub-national data provided by CANDID. First, we describe how poliomyelitis incidence was strongly and consistently seasonal from 1933 to 1963 and that the yearly peaks were synchronous across provinces. This pronounced synchrony contrasts with patterns in the United States [11], but can be interpreted within the context of a broader latitudinal gradient in the timing of seasonal epidemic peaks across North America. Second, the well-studied apparent resurgence of whooping cough in the 1990s showed significant regional variation. While the territories, prairie provinces, and Québec experienced clear increases in incidence, this pattern was not evident in British Columbia, Ontario, and the Atlantic region. We use a simple graphical approach in these examples, illustrating how these data can inform fundamental questions in epidemiology. Formal statistical analyses that dig deeper into specific questions will follow in subsequent publications. The Canadian Disease Incidence Dataset (CANDID) has the potential to drive many studies by the broader public health research community.

## Materials and methods

### Data sources

We began searching for Canadian historical infectious disease notification data in 2000. We focused on sub-annual and sub-national data collected through the surveillance programs described in Section A in S1 Appendix, particularly incidence over entire provinces and territories, since finer spatial resolution data were rare. We compiled provincial and territorial population data [12–14] to compute comparable incidence rates per 100,000. We acquired data in any of three formats: (1) paper hard copies, (2) digitally produced PDF files, or (3) spreadsheets (including CSV files). We scanned all of the hard copies. We found that Optical Character Recognition (OCR) was unable to convert scans into digital spreadsheets with sufficient accuracy, so we entered the data manually (see the Data entry section). We used PDF extraction tools [15] to avoid manual entry of digitally-produced PDF pages.

### Data entry

We manually entered the information in scans into replica Excel spreadsheets (i.e., digital spreadsheets in which the layout of each spreadsheet matches the original), to facilitate comparing reproductions with their sources (Fig 1). The Ontario Ministry of Health data were entered before our comprehensive effort and were only partially entered as replicas (they are the sole exception). Reading scans and interpreting handwritten sources (Fig 2) was often a slow, and occasionally error-prone, process. Where available in these sources, we also entered reported annual and national data to support later validation against marginal totals (see the Quality control section). When we encountered unclear numbers in the source material, we recorded our initial guesses using a predetermined format described in Section E in S1 Appendix, so that they could be processed systematically. These guesses were revisited and refined as needed when validation procedures, which compared sums of incidence values with marginal totals reported in both CANDID sources and the PHAC portal [10], revealed discrepancies. Scripted data preparation pipelines (see the Data preparation section) facilitated updates throughout this process. We provide details on our data entry process in Section B in S1 Appendix.

Top panel (typewritten source document):

| No. | Disease (For rare diseases, see page 9) | CANADA | | | Cumulative Total – Total cumulatif | | |
|---|---|---|---|---|---|---|---|
| | | Report Week – Semaine du rapport | Previous Week – Semaine précédente | Median 1960-1964 Médiane | 1965 | 1964 | Median – Médiane 1960-1964 |
| 1 | Brucellosis (Undulant fever) (044) | – | 1 | 1 | – | 2 | 1 |
| 2 | Diarrhoea of the newborn, epidemic (764) | 2 | 2 | 1 | 2 | 1 | 1 |
| 3 | Diphtheria (055) | – | – | 1 | – | – | 1 |
| 4 | Dysentery: | 16 | 17$^r$ | 39 | 16 | 55 | 39 |
| 5 | (a) Amoebic (046) | – | – | – | – | – | – |
| 6 | (b) Bacillary (045) | 10 | 17$^r$ | 21 | 10 | 26 | 21 |
| 7 | (c) Other and unspecified (048) | 6 | – | 18 | 6 | 29 | 18 |
| 8 | Encephalitis, infectious (082.0) | – | – | – | – | – | – |
| 9 | Food poisoning: | 14 | 11 | 11 | 14 | 20 | 11 |
| 10 | (a) Staphylococcus intoxication (049.0) | – | 7 | – | – | – | – |
| 11 | (b) Salmonella with food as vehicle of infection. (042.1) | 2 | 4 | 11 | 2 | 20 | 11 |
| 12 | (c) Unspecified (049.2) | 12 | – | – | 12 | – | – |
| 13 | Hepatitis, infectious (including serum hepatitis). (092,N998.5) | 118 | 132 | 140 | 118 | 111 | 140 |

Bottom panel (Microsoft Excel replica):

| | A | B | C | D | E | F | G | H |
|---|---|---|---|---|---|---|---|---|
| 3 | | | CANADA | | | | | |
| 4 | | | | | | Cumulative Total | | |
| 5 | No. | Disease (For rare diseases, see page 9) | Report Week - Semaine du rapport | Previous Week - Semaine précédente | Median 1960-1964 Médiane | 1965 | 1964 | Median - Médiane 1960-1964 |
| 6 | 1 | Brucellosis (Undulant fever) (044) | - | 1 | 1 | - | 2 | 1 |
| 7 | 2 | Diarrhoea of the newborn, epidemic (764) | 2 | 2 | 1 | 2 | 1 | 1 |
| 8 | 3 | Diphtheria (055) | - | - | 1 | - | - | 1 |
| 9 | 4 | Dysentery | 16 | 17$^r$ | 39 | 16 | 55 | 39 |
| 10 | 5 | (a) Amoebic (046) | - | - | - | - | - | - |
| 11 | 6 | (b) Bacillary (045) | 10 | 17$^r$ | 21 | 10 | 26 | 21 |
| 12 | 7 | (c) Other and unspecified (048) | 6 | - | 18 | 6 | 29 | 18 |
| 13 | 8 | Encephalitis, infectious [1] (082.0) | - | - | - | - | - | - |
| 14 | 9 | Food poisoning: | 14 | 11 | 11 | 14 | 20 | 11 |
| 15 | 10 | (a) Staphylococcus intoxication (049.0) | - | 7 | - | - | - | - |
| 16 | 11 | (b) Salmonella with food as vehicle of infection. (042.1) | 2 | 4 | 11 | 2 | 20 | 11 |
| 17 | 12 | (c) Unspecified (049.2) | 12 | - | - | 12 | - | - |
| 18 | 13 | Hepatitis, infectious (including serum hepatitis) (092,N988.5) | 118 | 132 | 140 | 118 | 111 | 140 |

**Fig 1**. **Example of a typewritten source document prepared using a typewriter.** The top panel shows a scan of part of this source document, and the bottom panel shows our replica in Microsoft Excel of this same part.

## Data preparation

We developed open-source [16] pipelines that convert replica spreadsheets (Fig 1) into CSV files, using common variables to combine data from different sources (details in Sections D-G in S1 Appendix). The focal variable is the number of new cases of a specific disease reported in a specific location over a specific time period. CANDID consists of three CSV files, ranging from a minimally processed file offering maximum flexibility in data preparation to a heavily processed file prioritizing convenience; each CSV file corresponds to a stage in the data processing pipeline (Fig A in S1 Appendix).

The unharmonized file preserves the raw, digitized data, giving researchers the freedom to apply their own data preparation methods. Descriptors in the unharmonized file use original names (e.g., "infantile paralysis" for poliomyelitis before

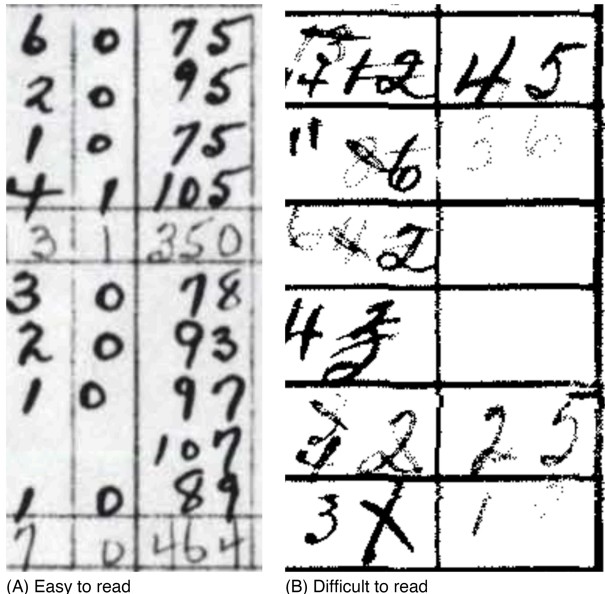

(A) Easy to read (B) Difficult to read

**Fig 2**. **Examples of handwritten data**. Most handwritten hard copies, such as the 1939 erysipelas and gonococcal data from the Ontario Ministry of Health (A), are easy to read. Others, like the 1955 poliomyelitis data from Statistics Canada (B), are difficult to read, posing challenges for digitization.

1924) to minimize historical information loss [17]. The harmonized file removes low quality data, aggregates some municipal data to provincial levels, and adds harmonized location and disease descriptors that simplify the combination of data from different sources (e.g., poliomyelitis whenever infantile paralysis is reported). Harmonized CSV files are convenient for querying diseases, provinces, and time periods, but they cannot be used directly for analysis due to overlapping incidence values. In data science terms, the harmonized data are not normalized [18,19]. For example, the harmonized data include total polio incidence alongside separate values for polio with and without paralysis (see Section G in S1 Appendix for more examples). Such overlapping data are useful for quality control (which we explain in the Quality control section), but data analysis requires removing overlaps to prevent double-counting cases. The normalized file does not contain overlapping incidence values, enabling aggregation without double-counting. When deciding which overlapping records to remove, we generally retain the finest resolution. Continuing our example, we would remove total polio incidence and retain separate records for polio with and without paralysis (see Section G in S1 Appendix for details on removal criteria). For convenience when computing incidence rates, we joined provincial population sizes to the normalized data. All figures in this paper are based on the normalized file. Researchers who wish to make different harmonization or normalization choices can use the upstream files (see Section G in S1 Appendix for details).

### Data provenance

Each record can be traced back to the relevant original scan, replica spreadsheet, and/or script used to produce it, using information in the CSV files (Section H in S1 Appendix). We follow research data management practices by distributing DataCite [20] (version 4.3) metadata with each dataset in the archive. These metadata will make it easier to deposit future versions of CANDID into a research data repository, which we plan to do.

### Quality control

We compared sums of incidence values with marginal totals reported both in CANDID sources and in the PHAC portal [10]. Discrepancies suggest possible data-entry or scripting errors. We investigated discrepancies and fixed those that

appeared to be due to digitization error. These investigations were simplified using our open data provenance tools (see the Data provenance section) and digitized data source replicas (Fig 1). We provide full detail on quality control in Section I in S1 Appendix.

## Results

As of October 2025, CANDID is based on 12 sources (Table 1). Section C in S1 Appendix provides information on how to access the resulting 1,631,380 unharmonized, 1,186,778 harmonized, and 934,010 normalized incidence values for the provinces and territories of Canada (Fig 3).

Figs 4, 5, and 6 list the 139 diseases that appear in the normalized dataset, and highlight the time periods in which weekly, monthly, or quarterly incidence data were found in each province or territory. Within each disease, vertical place-ment indicates data availability for a particular province or territory, ordered approximately clockwise as indicated on the

**Table 1**. **Data sources**. The Frequency column gives the shortest period over which incidence counts were reported for all diseases and locations in the source (if not all disease-location combinations have the shortest period, multiple frequencies are given). Sources that include handwritten data are indicated in the Received As column. Details on these sources are provided in Section A in S1 Appendix.

| Years | Provinces | Frequency | Organization | Received As |
|---|---|---|---|---|
| 1903-1947 | Ontario | monthly | Ontario Ministry of Health | Hard copy |
| 1910, 1921-1927 | Saskatchewan | monthly | Saskatchewan Department of Public Health | Hard copy |
| 1915-1925 | Québec | monthly | Québec Ministry of Health and Social Services | Hard copy |
| 1924-1955 | All | weekly, monthly | Statistics Canada | Hard copy (with handwriting) |
| 1939-1989 | Ontario | weekly | Ontario Ministry of Health | Hard copy (with handwriting) |
| 1956-1978 | All | weekly | Statistics Canada | Hard copy |
| 1979-1989 | All | 4-weekly | Statistics Canada | Hard copy |
| 1990-2001 | All | monthly, quarterly | Health Canada | Hard copy |
| 1990-2021 | Ontario | weekly | Public Health Ontario | Digital spreadsheet |
| 2001-2006 | All | quarterly | Canada Communicable Disease Report | PDF |
| 2004-2017 | Manitoba | monthly | Public Health Manitoba | PDF |
| 2004-2019 | Alberta | weekly | Alberta Health | Digital spreadsheet |

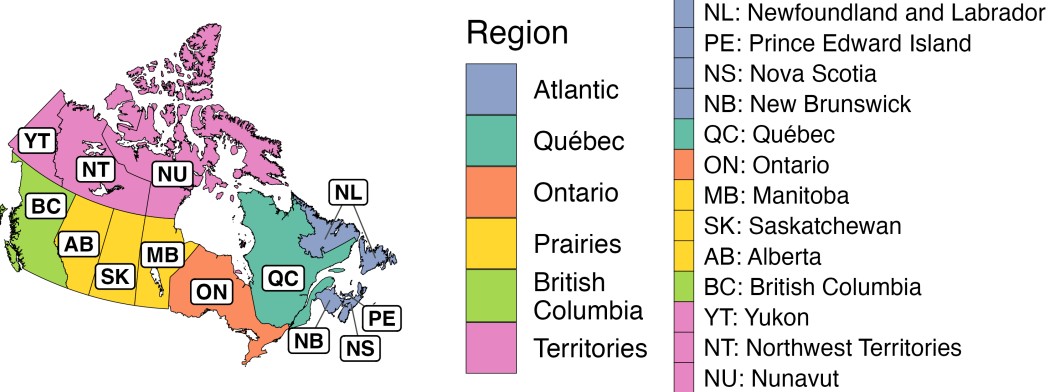

Region

| | |
|---|---|
| ■ | Atlantic |
| ■ | Québec |
| ■ | Ontario |
| ■ | Prairies |
| ■ | British Columbia |
| ■ | Territories |

NL: Newfoundland and Labrador
PE: Prince Edward Island
NS: Nova Scotia
NB: New Brunswick
QC: Québec
ON: Ontario
MB: Manitoba
SK: Saskatchewan
AB: Alberta
BC: British Columbia
YT: Yukon
NT: Northwest Territories
NU: Nunavut

**Fig 3**. **Map of Canada with provinces and territories labelled**. The map is used to show the order of provinces and territories in Figs 4, 5, and 6, and the order of provinces in the poliomyelitis example (see the Results section). The map also illustrates the grouping of provinces and territories into regions that are used in the whooping cough example (see the Results section). This map was prepared using the `rnaturalearth R` package [21], which utilized the following public domain shapefile: https://github.com/nvkelso/natural-earth-vector/blob/e08a35c801ac729401e2de9f5eef206031e6a284/10m_cultural/ne_10m_admin_1_states_provinces.shp.

map in Fig 3. The case numbers for many of these diseases are aggregated from 315 "sub-diseases" (Section F in S1 Appendix), harmonizing 929 unique historical name variants. The stratification of each disease into sub-diseases varied across sources (details in Section G in S1 Appendix).

### Examples

**Poliomyelitis cases peaked at the same time each year across all provinces.** Poliomyelitis incidence was strongly seasonal (upper panel, Fig 7), with national peaks consistently occurring between week 31 and 40 each year from 1933 to 1963 after which the cycles disappeared. These annual cycles were synchronous across provinces, with peak incidence occurring around the same time in each (provincial panels, Fig 7). All national peaks fell between August and October, and most provincial peaks followed this pattern, with only a few outlying province-year combinations (Fig I in S1 Appendix). Identifying such spatial synchrony requires incidence data at sub-annual and sub-national scales.

**Regional differences in whooping cough incidence.** Aggregating provincial whooping cough data to the national level (Fig 8) reveals a pattern consistent with previous analyses that lacked provincial data [22]. One feature of this pattern is an apparent resurgence of whooping cough in the 1990s (highlighted in Fig 8). We find that this much-discussed resurgence (e.g., [22]) was not uniformly expressed across the country (Fig 8 bottom six panels), and is clearly apparent only in the territories, the prairies, and Québec. Throughout the 1990s, yearly cases per 100,000 peaked at 21 in Ontario, but in the territories the peak was 293. The original study [22] did not have the sub-national data required to explore these regional differences.

### Discussion

CANDID complements existing, easily accessible Canadian notification data. The Public Health Agency of Canada (PHAC) provides an online portal [10] (https://diseases.canada.ca/notifiable) with annual, national incidence counts often used in retrospective analyses (e.g., [22,24–28]). However, these data lack the detail needed to study outbreak patterns, seasonality, or geographic variation. Our sub-annual, sub-national data enable research on patterns of variation within years and across provinces in Canada, and facilitate comparisons with U. S. data [29]. While Public Health Ontario maintains a tool for authorized public health professionals to access monthly data since 2012 [30], our public archive includes weekly Ontario data (1990–2021) and extends back before 1924, including Ontario (1903), Saskatchewan (1910), and Québec (1915). By consolidating federal and provincial sources (Section A in S1 Appendix) into a standardized format (Sections D-G in S1 Appendix), we simplify integration of new data, enabling researchers to focus on analysis rather than curation.

Our project parallels Project Tycho [31], which curated weekly U.S. incidence data and whose impact is summarized by [29]. Unlike Tycho, we open-sourced our data preparation pipelines to enable community-driven quality improvements. These pipelines include scans of original documents, spreadsheet replicas, and scripts to convert them into tidy CSV files. To our knowledge, no other studies publish such replicas (see the Data entry section), which help identify and correct data-entry errors. Our open-science approach enables researchers to trace incidence counts back to original sources (Section H in S1 Appendix) and improve data quality over time.

A well-established feature of poliomyelitis is its seasonal pattern, with epidemics in North America typically occurring in summer. Ref [11] attributed earlier epidemics in southern U.S. states to higher transmission rates. Our data extend this latitudinal gradient northward: epidemics across Canada tended to occur later (August–October) than those in the U. S. (May–July). However, we find no substantial differences in timing among Canadian provinces from 1934 to 1960, despite large variation in climate.

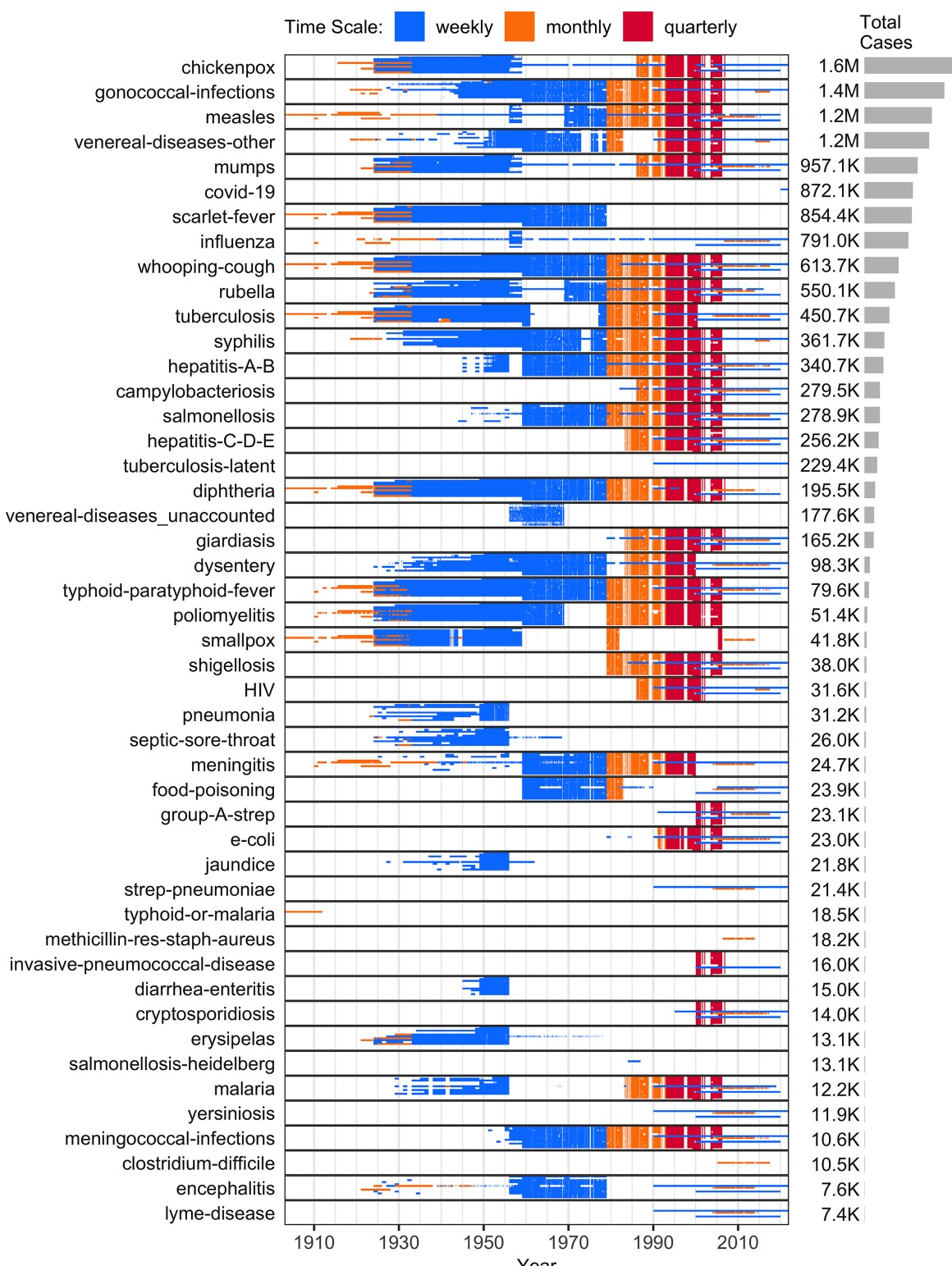

**Fig 4**. **Data availability for highly-reported diseases (top 47 by total reported cases)**. Moderately- and rarely-reported diseases appear in Figs 5 and 6. The diseases are ranked by the total number of cases (right panels), summed over all provinces for which data were obtained. The diseases

are ordered with the largest number of cases at the top. Each incidence value, including zeros, is shown as a tiny coloured rectangle. The y-axis labels identify the disease, while the rectangle's length along the x-axis represents the temporal extent. Colours indicate reporting frequency: weekly (blue), monthly (orange), and quarterly (red). Two-weekly data are shown in green but are not included in the legend because they are too few to see without zooming. Three-quarterly data are omitted, since they were reported only in 1997 and spanned most of that year (April-December 1997), providing little additional information on within-year variation. White spaces represent missing data (see Section E in S1 Appendix for details on the varied reasons for missing data). The vertical position within each disease denotes the province/territory, arranged roughly clockwise (east-to-west for provinces and then west-to-east for territories): NL-NS-PE-NB-QC-ON-MB-SK-AB-BC-YT-NT-NU (Fig 3). Thin horizontal patterns arise because provinces and territories differ in data availability and reporting frequency: when data are available for only some provinces, or when provinces and territories report at different time scales, the resulting variation appears as horizontal features due to the fixed top-to-bottom ordering of provinces and territories within each disease.

## Future directions

In this initial release of CANDID, we focused on sub-annual and sub-national disease incidence, with plans to expand further. First, we will include data stratified by age, sex, and municipality for available time ranges, provinces, and diseases. Age data, for example, can be critical for estimating the impact of vaccination programmes in childhood diseases (e.g., [32]). Second, we will extend the dataset's time range, disease coverage, and geographic detail as finer-scale or corrected data become available. Third, we will curate population-level information useful for epidemiological analyses, such as birth rates, mortality, vaccination, and school-term dates. Beyond expansion, we will address open questions in Canadian infectious disease history. For example, we will test whether regional differences in vaccination programmes could explain spatial variation in the size of the 1990s resurgence of whooping cough (Fig 8). Finally, we will explore AI-powered optical character recognition, using our archive as a training dataset to enhance its efficiency and accuracy.

## Logistical challenges

The process of assembling CANDID highlighted logistical obstacles that are likely to arise in other countries seeking to build comparable historical surveillance datasets. The first challenge was locating and obtaining source material. Historical records were dispersed among federal and provincial publications issued under evolving titles and formats, with many available only as hard copies in archives or libraries. Coordinated searches, correspondence with data stewards, and formal data requests to multiple agencies and libraries were required. These challenges are described in more detail in Section A in S1 Appendix.

A second challenge was manual data entry. Optical character recognition could not accurately extract numerical tables from scans, necessitating full manual transcription. Coordinating several data enterers required standardized templates, shared conventions for unclear entries, and frequent communication to ensure consistency across spreadsheets. Looking ahead, advances in artificial intelligence may mitigate some of these difficulties, as noted in our future plans (see the Future directions section), although many questions remain.

A third challenge was harmonizing information drawn from sources that differed widely in format and structure. Converting these diverse formats into consistent, machine-readable CSV files and resolving overlaps across locations, time scales, disease hierarchies, and data sources required careful sequencing of processing steps and version control across multiple repositories. These challenges are detailed in Sections D-G in S1 Appendix.

Finally, quality control involved iterative cross-checks across diseases, jurisdictions, and time scales. Addressing discrepancies meant revisiting earlier stages of data entry and processing, emphasizing the importance of reproducible pipelines. These challenges illustrate the logistical scale of constructing a comprehensive historical surveillance archive. By documenting our workflows and tools openly, we have worked for CANDID to serve as both a resource and a practical starting point for researchers undertaking similar digitization efforts elsewhere.

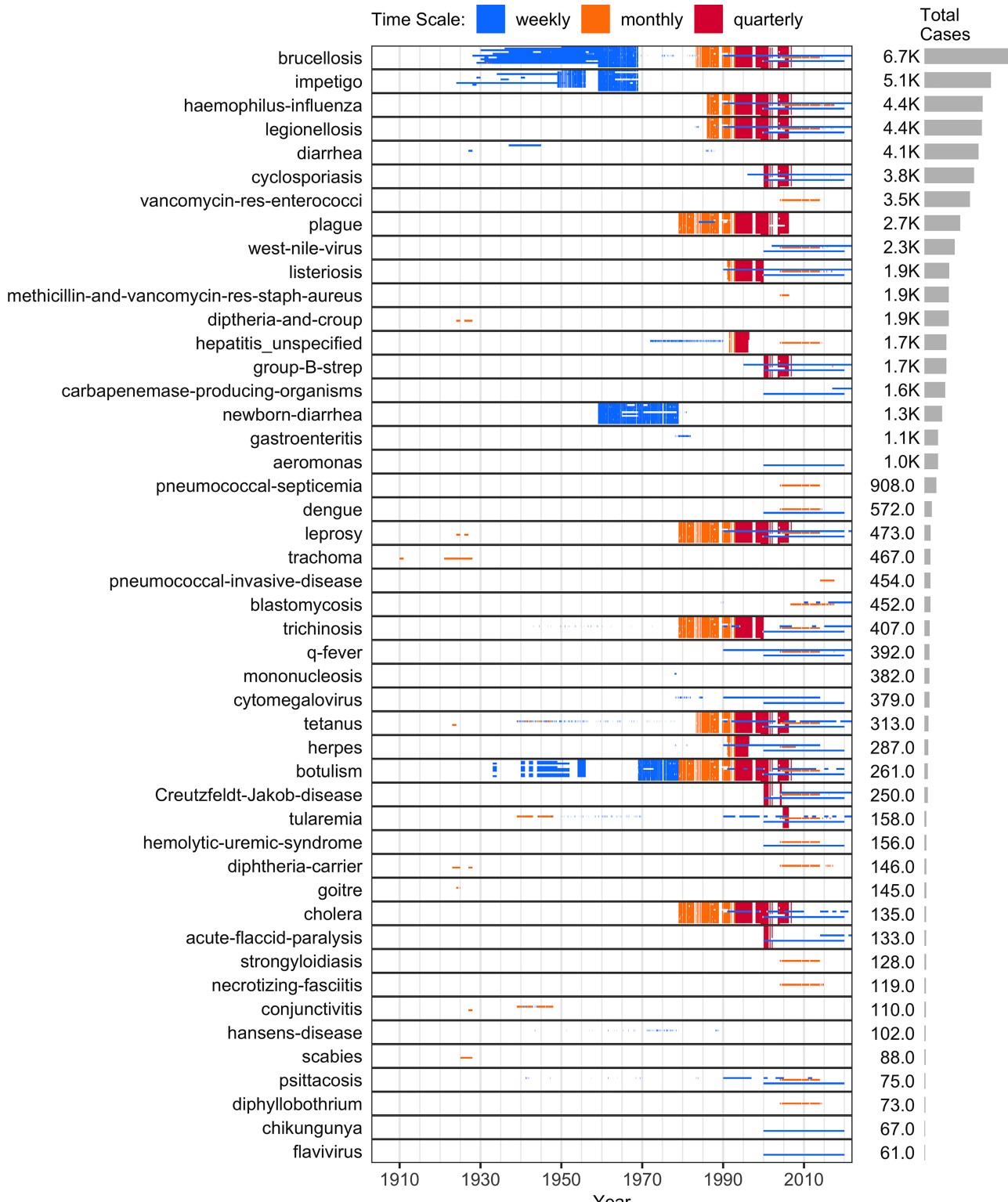

**Fig 5. Data availability for moderately-reported diseases (middle 47 by total reported cases).** Highly- and rarely-reported diseases appear in Figs 4 and 6. Please see the caption for Fig 4 for a full description of all of these plots.

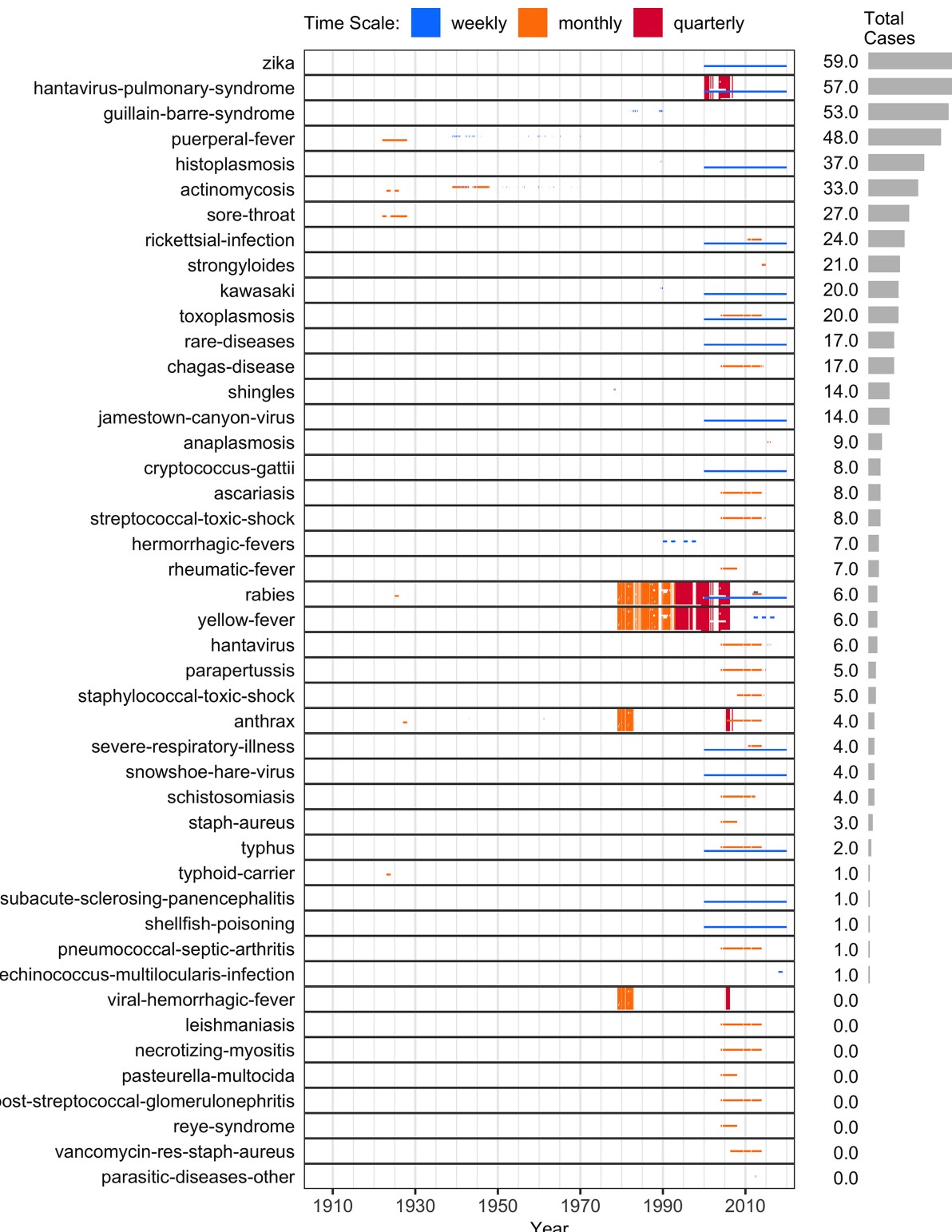

**Fig 6. Data availability for rarely reported diseases (bottom 45 by total reported cases).** Highly- and moderately-reported diseases appear in Figs 4 and 5. Please see the caption for Fig 4 for a full description of all of these plots.

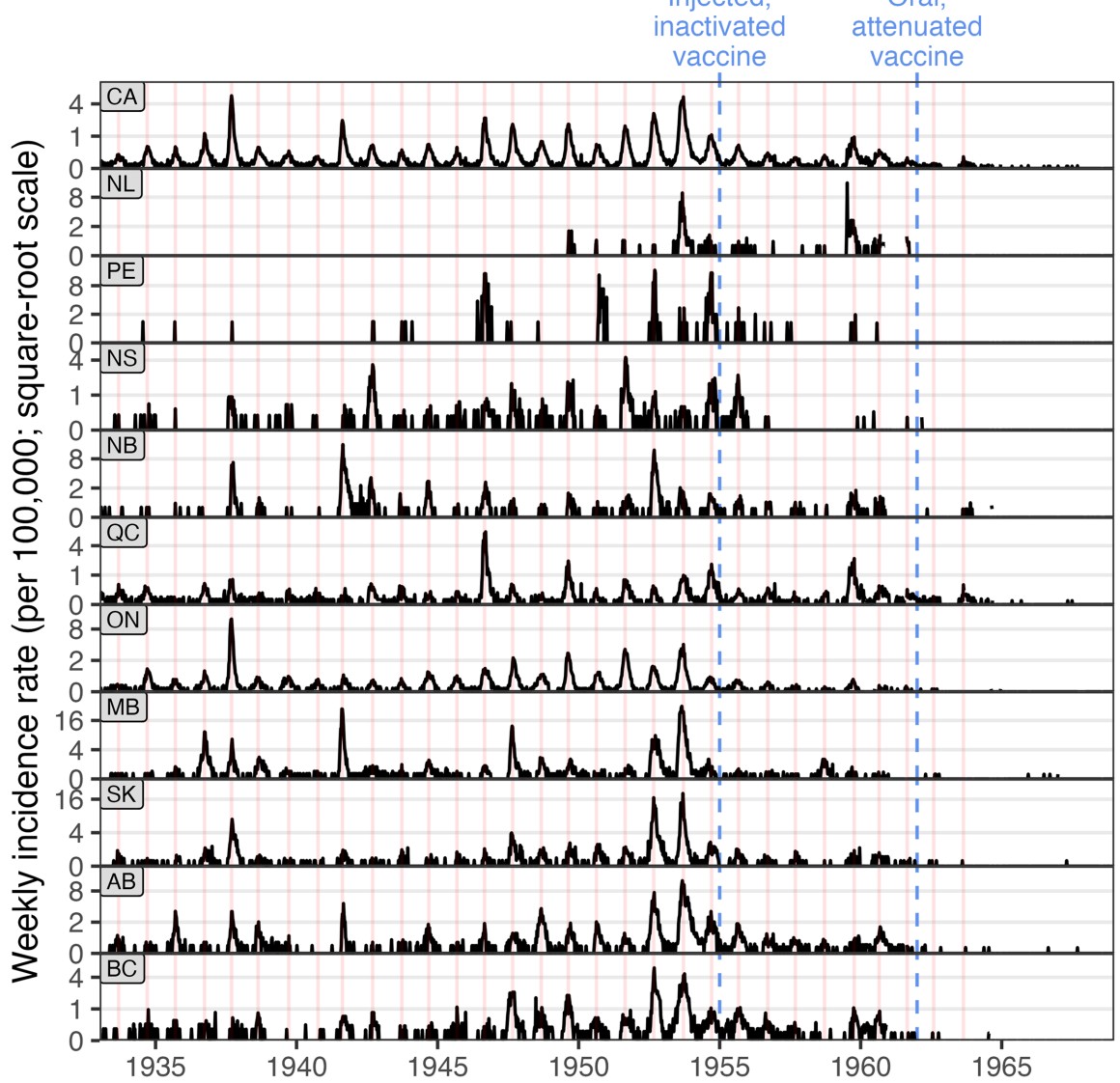

**Fig 7**. **Weekly poliomyelitis incidence from 1933 to 1968 (square root scale)**. The vertical lines do not indicate the start of each year but mark the week of peak national incidence (top panel) in years with more than 20 cases. Provincial peaks closely align with these national peaks, indicating strong spatial synchrony. This pattern could not have been detected with annual or national data. The introduction of two important vaccination programmes are shown as blue vertical dashed lines. We plot incidence rates per 100,000 to make incidence comparable across provinces and territories, and with other studies. Methods for producing this plot are described in Section J in S1 Appendix.

## Limitations

Under-reporting is a known limitation of surveillance data [31,33]. Correcting for it requires disease- and context-specific methods that rely on supplementary data (e.g., serological surveys, case-fatality ratios, demography) and modelling [34]. Given the number of diseases and years covered, such corrections are beyond our scope. Our contribution is to make data available in a form that supports future work, including efforts to address under-reporting. As one step in that direction, we distinguish true zero case counts from data that were not reported, based on information available in the original sources (see Section E in S1 Appendix).

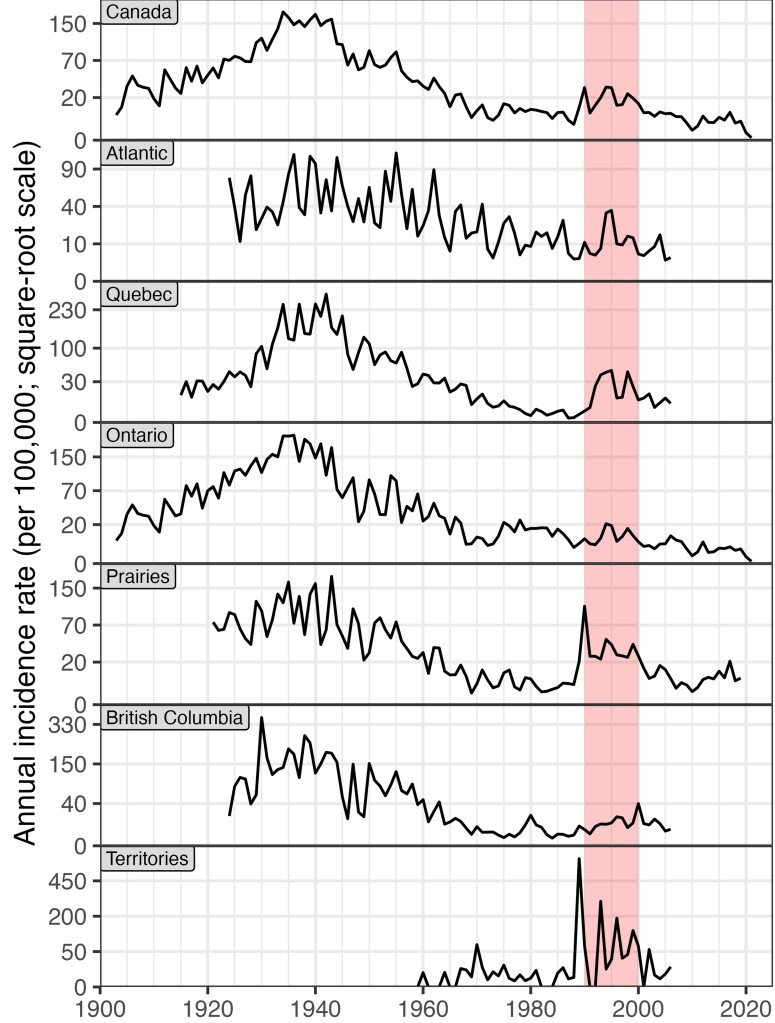

**Fig 8**. **Regional [23] differences in average annual whooping cough incidence in Canada over twelve decades (square root scale)**. The national data (top panel) are very similar to the first figure in a review based on different data sources [22]. The red region (1990–1999) highlights the first resurgence in national whooping cough incidence since widespread vaccination began in 1943. This plot shows that not all regions peaked in the 1990s, a pattern that could not have been detected with the national data used in the original study. Methods for producing this plot are described in Section K in S1 Appendix.

In addition to under-reporting, two other factors complicate the interpretation of these incidence data: evolving sub-disease hierarchies and inconsistent time scales. First, changes in how diseases are classified and reported over time affect comparability, as the level of aggregation can vary year to year (e.g., the normalized dataset includes 37 meningitis sub-diseases, with 1–15 reported in any given year; Figs C–F in S1 Appendix). Second, variation in reporting frequency (e.g., weekly, monthly, quarterly; Figs 4, 5, and 6) hinders the construction of evenly spaced time series, which are often needed for modelling and visualization.

Historical gaps in CANDID persist due to surveillance program changes (e.g., chickenpox was not notifiable from 1959–1985) and incomplete source coverage, particularly outside Québec, Ontario, and Saskatchewan before 1924. These gaps range from missing weeks (e.g., lost book pages) to illegible handwritten records (see the Quality control section). Although transcription and coding errors cannot be ruled out, we compared reported subtotals with their marginal

totals and corrected all identified discrepancies for our primary example diseases (whooping cough, poliomyelitis). By releasing open data pipelines, we aim to enable similar checks across all diseases and to support collective efforts to improve the completeness and reliability of CANDID over time (details in Section I in S1 Appendix).

## Conclusion

More than a century of infectious disease surveillance in Canada has produced a valuable record of epidemic patterns that has been largely unexploited, but can now be easily accessed. Comprehensive sub-annual and sub-national Canadian infectious disease incidence data have previously been unavailable. Similar data from other countries have been critical to establishing the foundations of epidemiological modelling and continue to push the field forward (e.g., [1–4,35–42]).

CANDID makes it possible to study variation in disease incidence within years and across provinces, with applications ranging from infectious disease research to broader interdisciplinary work. For example, it can be used to assess how incidence in Canada relates to socio-economic factors such as urbanization and wealth inequality. The dataset also supports public health planning by situating recent outbreaks and epidemics within their historical context.

In principle, it should be straightforward to keep the archive up to date, but doing so will require the cooperation of provincial and territorial public health agencies, which have released *less* data publicly since strictly digital data collection began in the 1990s. We contacted all these agencies but were able to obtain recent weekly data from only two provinces. While we recognize that agencies may face practical constraints, such as the staff time required to prepare and maintain public releases, we encourage Canadian governments to support routine public access to weekly, aggregated counts of infectious disease notifications.

## Supporting information

**S1 Appendix. Methodological details.** This supporting information details the historical and methodological framework for compiling and preparing Canadian notifiable infectious disease incidence data, and how to access them. It explains the sources and evolution of the Canadian Notifiable Disease Surveillance System (CNDSS), including federal and provincial publications, and describes the process of locating, digitizing, and cleaning these historical records. Data entry and quality control procedures are described, including internal consistency checks across time scales, locations, and disease hierarchies, as well as comparisons among data sources. Finally, the document outlines specific analytical methods for diseases like poliomyelitis and whooping cough.
(PDF)

## Acknowledgments

We are deeply thankful to Alberta Health and Public Health Ontario for providing us with recent weekly incidence data. Research assistants Jen Freeman, Frank Jin, Ronald Jin, and Steven Lee wrote code that we used during this project. Research assistants Jeanne Lin, Saul Widrich, Qinxian Zhu, Claire Lees, and Julia Maja entered some of the data. Research assistants Susan Marsh-Rollo, Maya Earn, Arielle Earn, and Elizabeth O'Meara found, organized, and scanned source documents. We thank Angie Fazone, who in 2001 helped DJDE locate handwritten weekly Ontario records (1939-1989) at the Ontario Ministry of Health, providing the first indication that the project was feasible. We appreciate the enthusiastic encouragement we received from Caroline Colijn, Michael Li and many other colleagues in the Canadian Network for Modelling Infectious Diseases (CANMOD).

## Author contributions

**Conceptualization:** David J. D. Earn, Steven C. Walker.

**Data curation:** David J. D. Earn, Gabrielle MacKinnon, Samara Manzin, Michael Roswell, Steve Cygu, Chyunfung Shi, Steven C. Walker.

**Formal analysis:** David J. D. Earn, Gabrielle MacKinnon, Samara Manzin, Michael Roswell, Steve Cygu, Benjamin M. Bolker, Jonathan Dushoff, Steven C. Walker.

**Funding acquisition:** David J. D. Earn, Benjamin M. Bolker, Jonathan Dushoff.

**Investigation:** David J. D. Earn, Gabrielle MacKinnon, Samara Manzin, Michael Roswell, Steve Cygu, Chyunfung Shi, Benjamin M. Bolker, Jonathan Dushoff, Steven C. Walker.

**Methodology:** David J. D. Earn, Gabrielle MacKinnon, Samara Manzin, Michael Roswell, Steve Cygu, Chyunfung Shi, Benjamin M. Bolker, Jonathan Dushoff, Steven C. Walker.

**Project administration:** David J. D. Earn.

**Resources:** David J. D. Earn, Steven C. Walker.

**Software:** David J. D. Earn, Gabrielle MacKinnon, Samara Manzin, Michael Roswell, Steve Cygu, Steven C. Walker.

**Supervision:** David J. D. Earn, Steven C. Walker.

**Validation:** David J. D. Earn, Gabrielle MacKinnon, Samara Manzin, Michael Roswell, Steve Cygu, Chyunfung Shi, Steven C. Walker.

**Visualization:** David J. D. Earn, Gabrielle MacKinnon, Samara Manzin, Michael Roswell, Steve Cygu, Steven C. Walker.

**Writing – original draft:** Steven C. Walker.

**Writing – review & editing:** David J. D. Earn, Gabrielle MacKinnon, Samara Manzin, Michael Roswell, Steve Cygu, Chyunfung Shi, Benjamin M. Bolker, Jonathan Dushoff, Steven C. Walker.

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
