## [Editor Report · Decision Letter 0]

14 Apr 2025

PGPH-D-25-00360

Over a Century of Infectious Disease Surveillance in Canada

Dear Dr. Walker,

Thank you for submitting your manuscript to PLOS Global Public Health. After careful consideration, we feel that it has merit but does not fully meet PLOS Global Public Health’s publication criteria as it currently stands. We have not been successful in securing reviewers for this manuscript and apologise for the delay. As the editor, I have reviewed the manuscript and invite you to submit a revised version of the manuscript that addresses the points raised during the review process.

The epidemiological analysis lacks novelty, as the seasonal nature of polio is well-established.The paper is excessively long with sections that are difficult to comprehend, reducing its overall clarity and impact.There is insufficient discussion of NOID (Notifiable Disease) statutes across Canadian provinces and their evolution over time.The paper lacks adequate coverage of zero reporting policies and reporting efficiency analyses.Provincial abbreviations need clarification; a map would be beneficial for readers.The figures have substantial presentation issues:Figure 1's caption is insufficient and the figure is difficult to interpret. Colors should be referenced rather than wavelengths, and the horizontal lines are unexplained.Figure 2 uses an inappropriate square root scale (a logarithmic scale would be more suitable) and should display absolute numbers rather than rates.The inadequate horizontal scale in Figure 2 fails to identify the month of peak occurrence.

A more concise paper focusing on the history of disease notifications in Canada, accompanied by an announcement of the database's availability, with the current appendix material serving as supplementary information for interested researchers.

We hope these comments prove helpful for the further development of your research.

If you choose to proceed, please submit your revised manuscript by May 29 2025 11:59PM. If you will need more time than this to complete your revisions, please reply to this message or contact the journal office at globalpubhealth@plos.org. Please include the following items when submitting your revised manuscript:

A rebuttal letter that responds to each point raised by the editor. You should upload this letter as a separate file labeled 'Response to Reviewers'.A marked-up copy of your manuscript that highlights changes made to the original version. You should upload this as a separate file labeled 'Revised Manuscript with Track Changes'.An unmarked version of your revised paper without tracked changes. You should upload this as a separate file labeled 'Manuscript'.

We look forward to receiving your revised manuscript.

Kind regards,

Palwasha Yousafzai Khan, MBBCh, MSc, PhD

Academic Editor

Journal Requirements:

1. We ask that a manuscript source file is provided at Revision. Please upload your manuscript file as a .doc, .docx, .rtf or .tex.
---

## [Decision Letter · Decision Letter 1]

1 Sep 2025

PGPH-D-25-00360R1

Over a Century of Infectious Disease Surveillance in Canada

Dear Dr. Walker,

Thank you for submitting your manuscript to PLOS Global Public Health. After careful consideration, we feel that it has merit but does not fully meet PLOS Global Public Health’s publication criteria as it currently stands. Therefore, we invite you to submit a revised version of the manuscript that addresses the points raised during the review process.

Good to see the effort of the authors on improving the initial manuscript, we understand the manuscript took a bit more time in review process and the reviewers are having a bit diverse reports although the study is found interesting. I suggest authors for go for another round of minor but essential revision for further process. 

We look forward to receiving your revised manuscript.

Kind regards,

Sheikh Taslim Ali, M.Sc., Ph.D.

Academic Editor

Journal Requirements:

Additional Editor Comments (if provided):

Reviewers' comments:

Reviewer's Responses to Questions

**Comments to the Author**

1. If the authors have adequately addressed your comments raised in a previous round of review and you feel that this manuscript is now acceptable for publication, you may indicate that here to bypass the “Comments to the Author” section, enter your conflict of interest statement in the “Confidential to Editor” section, and submit your "Accept" recommendation.

Reviewer #1: All comments have been addressed

Reviewer #2: (No Response)

Reviewer #3: (No Response)

2. Does this manuscript meet PLOS Global Public Health’s publication criteria? Is the manuscript technically sound, and do the data support the conclusions? The manuscript must describe methodologically and ethically rigorous research with conclusions that are appropriately drawn based on the data presented.

Reviewer #1: Yes

Reviewer #2: Partly

Reviewer #3: Partly

3. Has the statistical analysis been performed appropriately and rigorously?

Reviewer #1: Yes

Reviewer #2: N/A

Reviewer #3: Yes

4. Have the authors made all data underlying the findings in their manuscript fully available (please refer to the Data Availability Statement at the start of the manuscript PDF file)?

Reviewer #1: Yes

Reviewer #2: Yes

Reviewer #3: Yes

5. Is the manuscript presented in an intelligible fashion and written in standard English?

Reviewer #1: Yes

Reviewer #2: Yes

Reviewer #3: Yes

6. Review Comments to the Author

Reviewer #1: This manuscript presented a valuable work that shows the establishment of a comprehensive and accessible dataset of historical record of Canadian's notifiable disease incidence is greatly helpful for epidemiological studies. I do appreciate this hard work and acknowledge its value, and I think the comments from the previous reviewer have been properly responded in this revised version. I only have some very slight comments listed below:

1. In my opinion, this manuscript is emphasizing the importance of setting up a comprehensive and high resolution dataset of notifiable disease incidence record, it might be better if the authors could indicate that from the title.

2. In discussion section, future directions, the authors may also discuss how this dataset could help in interdisciplinary studies. For example, incorporate with economy and urbanization studies, how the change patterns of infectious disease incidence in Canada were related with the progress of urbanization, should also be worth studying.

3. At the end of subsection 3.4 Conclusion, instead of urging the government to require public health agencies to release higher resolution data as soon as possible, the authors might use a gentle wording to encourage the government to do that, and may also add up a bit more brief discussion about some potential obstacles that might hinder the public health agencies to release such data.

Reviewer #2: This seems like a revised submission. Nevertheless, I commend the authors for the curation of such an important historical data. While an impressive and potentially time-consuming piece of work, there are some ambiguities that need clarification for a better understanding of what was done and to make the reporting clearer and more useful to researchers and other stakeholders. The PHAC, which I believe should have the entire curated data; albeit, potentially in typewritten/handwritten hard copies, as the authors suggested, provides summaries of the data from 1924-2022, but the authors have assembled incidence values from 1903-2021, which means an additional data from 21 years before 1924 and exclusion of 2022 data. It would be helpful to explain this disparity in the dates, particularly of the earliest available data because there is likely a reason why PHAC limited their data summaries to from 1924. There is also the need for the readers and potential users of the data to understand how the data differs from the summaries provided via PHAC because there are likely differences, however subtle they may be (for example, perhaps, PHAC provides intra-annual data summaries versus the sub-annual data, which is the focus of this work). It is worth mentioning that transparency, details, and full declaration are the key in data curation. These said, it would be helpful if the authors could also address the following concerns:

1. The terms intra-annual and sub-annual are of course timescales within a year but are nuanced and could create conflicts in the minds of some of the readers. As far as I am aware, intra-annual is a more precise term when describing events or variations in events happening within a year, while sub-annual is a broader term that encompasses any tracking of events less than a year. As such, it would be helpful to have these clarified at their first mention, and for consistency in their use throughout the study reporting.

2. The term sub-national needs to be clarified for less ambiguity. Perhaps, the authors are referring to provinces/territories, or maybe to regions, for example, the Prairies, Atlantic Canada etc.

3. The statement “We developed open-source pipelines to harmonize these spreadsheets into CSV files that blend data across sources” is unlikely to make sense to many, so needs clarifying/simplifying.

4. This study was conducted systematically, and therefore, could be likened to a systematic evidence review. As such, and for full transparency, the authors should consider adding more information to the methods to explain who (and how many) conducted literature/data search and how potential errors were mitigated, who (and how many) conducted data entry and what is meant by the statement “We took an iterative approach to resolving unclear numbers by ‘guessing and then refining our guesses’ if necessary”, who (and how many) conducted data preparation, and the necessary quality checks/assessments that were made to mitigate errors.

5. The authors reported that “the focus was on sub-annual and sub-national data and that removing redundancy yielded 934,010 weekly, monthly, or quarterly incidence values broken down by province/territory, containing 139 diseases.” It would be helpful to report what made an incidence value redundant, and why the focus was on sub-annual data that is clearly not truly representative of a year’s incidence report rather than the more appropriate intra-annual data that covers incidence in a full year. Moreover, you cannot accurately explore variability in events in a year or describe seasonality of an event, as the authors have claimed, without having considered the event incidence throughout a full year, in this case, intra-annual and not sub-annual data. Further, does 934,010 refer to week, month, or quarter? The sentence is muddled up, so needs clarifying.

6. Perhaps I am missing something, but I do not understand the statement “Our sub-annual, sub-national data enable research on intra-annual and inter-provincial patterns in Canada and facilitate comparisons with U.S. data”. How can sub-annual inform intra-annual, when intra-annual encompasses a whole year (meaning that it includes sub-annual) whereas sub-annual includes just a portion of a year?

7. Although trivial, you do not start a sentence with year in digits – so, instead of “2024 marked the 100th anniversary since Canada ……”, the authors should consider revising to “The year 2024 marked the….”. Similar revision should be made wherever else such a reporting may have been made.

8. Finally, if at all useful, the authors could refer to this study on whooping cough from Manitoba (https://pubmed.ncbi.nlm.nih.gov/35045177/).

Reviewer #3: This is an interesting study detailing the creation of a dataset (CANDID) that can be used for epidemiology and public health. Below, I provide some comments that can help in strengthening the manuscript.

Minor comments

• Could the authors state and clarify the objectives of the study? From the introduction, it seems that the purpose of the study was to create a dataset that records the incidence of diseases in Canada over time.

• Since the manuscript has been sent to a global health journal, could the authors mention how this methodology can be applied to international settings, where most of the data is in hard copies?

• The authors have mentioned the limitations of the study; however, it will be worthwhile to state the logistical and other difficulties faced during the collection of data. This would be helpful for researchers in other countries if they want to create similar datasets.

7. PLOS authors have the option to publish the peer review history of their article (what does this mean?). If published, this will include your full peer review and any attached files.

**Do you want your identity to be public for this peer review?** For information about this choice, including consent withdrawal, please see our Privacy Policy.

Reviewer #1: No

Reviewer #2: No

Reviewer #3: No

---

## [Decision Letter · Decision Letter 2]

12 Nov 2025

A Century of Weekly Notifiable Disease Incidence Data by Province in Canada

PGPH-D-25-00360R2

Dear Dr Walker,

We are pleased to inform you that your manuscript 'A Century of Weekly Notifiable Disease Incidence Data by Province in Canada' has been provisionally accepted for publication in PLOS Global Public Health.

Best regards,

Sheikh Taslim Ali, M.Sc., Ph.D.

Academic Editor

Authors revised the manuscrpt substantially addressing the issues raised by the reviewers.

Reviewer Comments (if any, and for reference):

Reviewer's Responses to Questions

**Comments to the Author**

1. If the authors have adequately addressed your comments raised in a previous round of review and you feel that this manuscript is now acceptable for publication, you may indicate that here to bypass the “Comments to the Author” section, enter your conflict of interest statement in the “Confidential to Editor” section, and submit your "Accept" recommendation.

Reviewer #1: All comments have been addressed

Reviewer #2: All comments have been addressed

Reviewer #3: All comments have been addressed

2. Does this manuscript meet PLOS Global Public Health’s publication criteria? Is the manuscript technically sound, and do the data support the conclusions? The manuscript must describe methodologically and ethically rigorous research with conclusions that are appropriately drawn based on the data presented.

Reviewer #1: Yes

Reviewer #2: Yes

Reviewer #3: Yes

3. Has the statistical analysis been performed appropriately and rigorously?

Reviewer #1: Yes

Reviewer #2: N/A

Reviewer #3: Yes

4. Have the authors made all data underlying the findings in their manuscript fully available (please refer to the Data Availability Statement at the start of the manuscript PDF file)?

Reviewer #1: Yes

Reviewer #2: Yes

Reviewer #3: Yes

5. Is the manuscript presented in an intelligible fashion and written in standard English?

Reviewer #1: Yes

Reviewer #2: Yes

Reviewer #3: Yes

6. Review Comments to the Author

Reviewer #1: The authors have addressed all of my comments, and I think this is a valuable work to be published and useful for future studies.

Reviewer #2: The authors have responded adequately to all my comments. The reporting of their study is far clearer that it was, and I can now read the manuscript with ease. I highly commend them for this substantial contribution to, not only the evidence base, but also to infectious diseases research in Canada. These said, it would be helpful to know whether access to CANDID is free and without any ethics requirements. Clarifying this would be helpful to the readers and potential users of the database. Further, considering the extensive revisions that the authors have made, I would advise a careful proofreading of this manuscript before publication to reduce the risk of the need for corrections, or retractions afterwards.

Reviewer #3: The authors have addressed all of my comments.

7. PLOS authors have the option to publish the peer review history of their article (what does this mean?). If published, this will include your full peer review and any attached files.

**Do you want your identity to be public for this peer review?** For information about this choice, including consent withdrawal, please see our Privacy Policy.

Reviewer #1: No

Reviewer #2: No

Reviewer #3: No
